# Precision Phenotyping of Wild Rocket (*Diplotaxis tenuifolia*) to Determine Morpho-Physiological Responses under Increasing Drought Stress Levels Using the PlantEye Multispectral 3D System

Pasquale Tripodi [1,*], Cono Vincenzo [1,2], Accursio Venezia [1], Annalisa Cocozza [1,2] and Catello Pane [1]

1    Research Centre for Vegetable and Ornamental Crops, Council for Agricultural Research and Economics (CREA), Via Cavalleggeri 51, 84098 Pontecagnano Faiano, Italy; cono.vincenzo@crea.gov.it (C.V.); accursio.venezia@crea.gov.it (A.V.); annalisa.cocozza@crea.gov.it (A.C.); catello.pane@crea.gov.it (C.P.)
2    Department of Agricultural Sciences, University of Naples Federico II, Via Università 100, 80055 Portici, Italy
*    Correspondence: pasquale.tripodi@crea.gov.it; Tel.: +39-089-386217

**Abstract:** The PlantEye multispectral scanner is an optoelectrical sensor automatically applied to a mechatronic platform that allows the non-destructive, accurate, and high-throughput detection of morphological and physiological plant parameters. In this study, we describe how the advanced phenotyping platform precisely assesses changes in plant architecture and growth parameters of wild rocket salad (*Diplotaxis tenuifolia* L. [DC.]) under drought stress conditions. Four different irrigation supply levels from moderate to severe, required to keep 100, 70, 50, and 30% of the water-holding capacity, were adopted. Growth rate and plant architecture were recorded through the digital measure of biomass, leaf area, Canopy Light Penetration Depth, five convex hull traits, plant height, Surface Angle Average, and Voxel Volume Total. Vegetation color assessments included hue, lightness, and saturation. Vegetation and senescence indices were calculated from canopy reflectance in the red (620–645 nm), green (530–540 nm), blue (peak wavelength 460–485 nm), near-infrared (820–850 nm), and 3D laser (940 nm) ranges. The temperature, relative humidity, and solar radiation of the environment were also recorded. Overall, morphological parameters, color, multispectral data, and vegetation indices provided over 7200 data points through daily scans over three weeks of cultivation. Although a general decrease in growth parameters with increasing stress severity was observed, plants were able to maintain the same morpho-physiological performances as the control during the early growth stages, keeping both 70% and 50% of the total water-holding capacity. Among indices, the Normalized Differential Vegetation Index (NDVI) contributed the most to the differentiation between different stress levels during the cultivation cycle. Across the 3 weeks of growth, statistically significant differences were observed for all traits except for the Saturation Average. Comparisons with respect to the control highlighted the strong impact of drought stress on morphological plant traits. This study provided meaningful insights into the health status of wild rocket salad under increasing drought stress.

**Keywords:** high-throughput detection; three-dimensional scan; leafy vegetables; plant reflectance; water deficit

## 1. Introduction

Traditionally, manual methods have been used to gather plant phenotypic data. These approaches present several drawbacks: low efficiency as it is labor-intensive to detect thousands of data points, a subjective manner of analysis, risks of inefficiency, and inaccuracy of the information collected [1]. Cutting-edge technologies based on spectral imaging provide an advanced application to dissect, in a non-destructive manner, plant phenotypes with high precision and throughput. This dataset presents the application of the PlantEye F500, a three-dimensional (3D) multispectral laser scanner able to capture over 20 plant parameters per

scan with high processivity when equipped on an automated phenotyping platform. Based on the 3D model obtained from spectral data in the RGB (red, green, blue) and near-infrared wavelengths, computer vision allows us to reconstruct the plant canopy and calculate some synthetic vegetational indices (VIs), thus determining the overall plant development and health status. This platform is being increasingly used in several studies and different cropping systems to investigate the physiological, stress tolerance, and nutritional-related traits in both open-field and protected environments [2–5]. Here, the PlantEye F500 has been used to monitor the growth and physiology of wild rocket (*Diplotaxis tenuifolia*) under water stress. Wild rocket salad is a herbaceous crop with an annual life cycle belonging to the Brassicaceae family, whose plant leaves are eaten as a vegetable, being appreciated by consumers for the typical pungent taste and spicy aroma [6]. It is also an invaluable source of health-beneficial compounds, being rich in vitamin C, flavonoids, and glucosinolates [7]. The species shows a certain tolerance to drought, a characteristic that makes it suitable for colonizing natural environments with a limited availability of water [8]. However, in agriculture systems, water management of the crop is necessary to modulate fertigation, pathogen susceptibility, nutritional and post-harvest quality traits, and water use efficiency [9], also in view of the general impacts of global warming. So, monitoring changes in plant morpho-physiological parameters under water stress is strategic for decision making (e.g., cultivar selection, implementation of agronomic protocols). Manual measurements of plants under drought stress conditions are laborious, not allowing for the precise dissection of changes in plant architecture and variations in physiological parameters. These limitations are even more drastic if different levels of water deficit occur during the growing cycle. Here, we describe the application of a precision phenotyping system with the aim of determining those traits that are primarily affected by a lack of irrigation in wild rocket. Additionally, this is the first attempt to use the PlantEye F500 to investigate the effects of drought stress on this crop. To the best of our knowledge, to date, this tool has not yet been used in *D. tenuifolia*. The approach presented here provides valuable information for researchers interested in the mechanisms of resistance/tolerance to drought stress in wild rocket toward the optimization of the water supply during its cultivation.

## 2. Materials and Methods

### 2.1. Plant Material and Experimental Details

The experiment was carried out in June 2023 in the glasshouse of CREA—Research Centre for Vegetable and Ornamental Crops, Pontecagnano Faiano, Italy (40°37′ N; 14°58′ E). *Diplotaxis tenuifolia* cv. Tricia, commonly known as wild rocket or Mediterranean wild rocket, was used for this study. Five just-emerged seedlings were transplanted into plastic pots (7 cm diameter and 100 mL vol.) filled with 120 g of peat, whose water-holding capacity was determined by the gravimetric method. An automated climate control system consisting of an evaporative fan cooling system and a shading screen was adopted. Two weeks after transplantation, plants were subjected to comparative treatments for 19 days, keeping growing substrates at a 100% water-holding capacity (leaching 20% of the applied water and electrical conductivity (EC) of the irrigation solution of 1.53 dS m$^{-1}$) (control, C); 70% water-holding capacity (30% less water than the control plants, 70); 50% water-holding capacity (50% less water than the control plants, 50); and 30% water-holding capacity (70% less water than the control plants, 30). The moisture pot content was measured every day at 9 a.m. Each pot was replenished daily with the evapotranspiration water lost. This research was executed in a randomized complete block design with five replications, each represented by a pot with 5 plants, for a total of 25 plants per treatment. No fertilization was applied.

### 2.2. Phenotyping Assessment

Plants were scanned using a PlantEye F500 multispectral 3D scanner (Phenospex, Heerlen, The Netherlands) [10]. The scanner moves horizontally and vertically (*X-Y* axis) on a gantry system positioned on the top of the plants, at 1.5 m (Figure 1), acquiring all

measures and reconstructing the whole plant canopy. The plants were scanned daily at 11:30 am. Each scan took 20 min to acquire information throughout the trials.

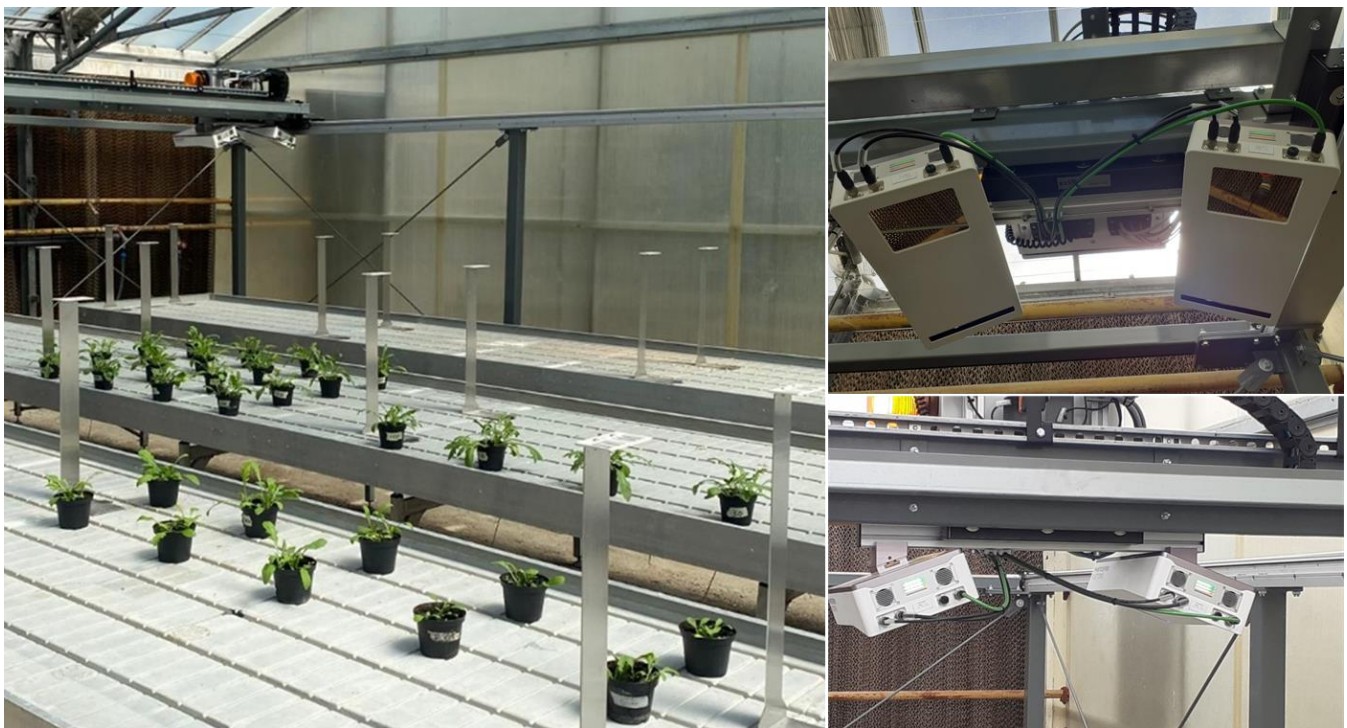

**Figure 1.** The Phenospex phenotyping platform of CREA—Research Centre for Vegetable and Ornamental Crops (Pontecagnano, Italy). On the left, the overview of the pot experiment is shown; vertical bars carry the coordinates (barcodes), allowing the 3D scanner to recognize the position of plants on the bench. On the right, details (front–bottom) of the PlantEye F500 dual scanner are shown.

PlantEye acquires spectral reflectance in four different wavelengths, each measured in one channel. Peak wavelengths (PWs) and spectral half width (SHW) for each channel are reported as follows: red (PW 620–645 nm; SHW 20 nm), green (PW 530–540 nm; SHW 80 nm), blue (PW 460–485 nm; SHW 20 nm), near-infrared (PW 820–850 nm; SHW 20 nm), and 3D laser (PW 940 nm). The PlantEye phenotyping system then automatically computes, by Hortcontrol v. 3.8 software [10], a diverse set of morphological plant parameters calculated from the 3D model of the plant (Table 1).

**Table 1.** List of measured traits calculated from the 3D model of the plant with the PlantEye 500 multispectral laser scan. Traits include (i) morphological parameters (13), color and multispectral reflectance measures (3), and spectral vegetation indices (4).

| Acronym | Trait | Unit of Measurement | Trait Type |
| --- | --- | --- | --- |
| LA3D | Three-Dimensional Leaf Area | mm$^2$ | Morphological parameters |
| CLPD | Canopy Light Penetration Depth | mm | |
| CHAC | Convex Hull Area Coverage | % | |
| CHA | Convex Hull Area | mm$^2$ | |
| CHAR | Convex Hull Aspect Ratio | index | |
| CHC | Convex Hull Circumference | mm | |
| CHMW | Convex Hull Maximum Width | mm | |
| DB | Digital Biomass | mm$^3$ | |
| PHA | Plant Height Averaged | mm | |
| PHM | Plant Height Max | mm | |
| PLA | Projected Leaf Area | mm$^2$ | |

**Table 1.** *Cont.*

| Acronym | Trait | Unit of Measurement | Trait Type |
|---|---|---|---|
| SAA | Surface Angle Average | A° | |
| VVT | Voxel Volume Total | mm³ | |
| HUE | Hue Average | Â° | Color and multispectral |
| LA | Lightness Average | % | |
| SA | Saturation Average | % | |
| NDVI | Normalized Differential Vegetation Index | index | Vegetation indices |
| NPCI * | Normalized Pigment Chlorophyll Index | index | |
| PSRI # | Plant Senescence Reflection Index | index | |
| GLI | Green Leaf Index Average. | index | |

* calculated as (RED − BLUE)/(RED + BLUE). # calculated as (RED − GREEN)/(NIR).

*2.3. Data Analysis*

All data were analyzed using the R statistical software v4.0.2 [11]. The ANOVA was performed to determine, for each trait studied, the significance of the different treatments in each week of cultivation. Average differences between treatments and the control were compared using the Dunnett tests. A $p = 0.05$ threshold was considered to indicate a statistically significant difference. The correlations among traits scored in each independent treatment were calculated from accession means using the *corrplot* R package v 4.4 [12]. The Pearson linear coefficients of correlation (r) were calculated between pairs of traits, and the significance of correlations was evaluated at $p < 0.05$. A principal component analysis (PCA) was carried out among accession means of 20 traits scored across three weeks for the different drought stress treatments, in order to determine the most effective traits in discriminating among accessions. PCA loading and score plots were drawn in R using the *FactoMineR* and *factoextra* packages [13,14]. The prediction ellipses with 95% confidence intervals were added to the PCA score plot.

**3. Results and Discussion**

During the period of cultivation, the internal temperature ranged from 19 °C (night) to 32 °C (day) with an average relative humidity of 72% ranging from 35% to 82% (Table 2: Data File 1). By scanning once a day, the PlantEye F500 allowed us to investigate all parameters in depth, thus determining the changes in morphological and physiological parameters in the different growing conditions. In total, we gathered 7200 phenotypic data points on both control and water-stressed plants from 8 June to 26 June 2023 (Table 2: Data File 2).

**Table 2.** Overview of Data Files reporting raw climatic and phenotyping data.

| Label | Name of Data File | Data Repository and DOI Identifier |
|---|---|---|
| Data File 1 | D. tenuifolia_Trial_Climate Datalogger | Figshare (https://doi.org/10.6084/m9.figshare.25201160, accessed on 6 May 2024) |
| Data File 2 | D_tenuifolia_Water_Stress_F500Phenotyping | Figshare (https://doi.org/10.6084/m9.figshare.25201172, accessed on 6 May 2024) |

The applied stresses highlighted substantial changes in the morphology and canopy of the plant (Figure 2). The Three-Dimensional Leaf Area consistently decreased with the incremental stress during the 3 weeks of this study. We observed how, in control conditions, LA3D increased from the first to the third week, while with both 50% and 30% water stress, during the third week, the lowest values were reached. This is justified by the reduced growth capacity of the plants and the onset of leaf necrosis mechanisms.

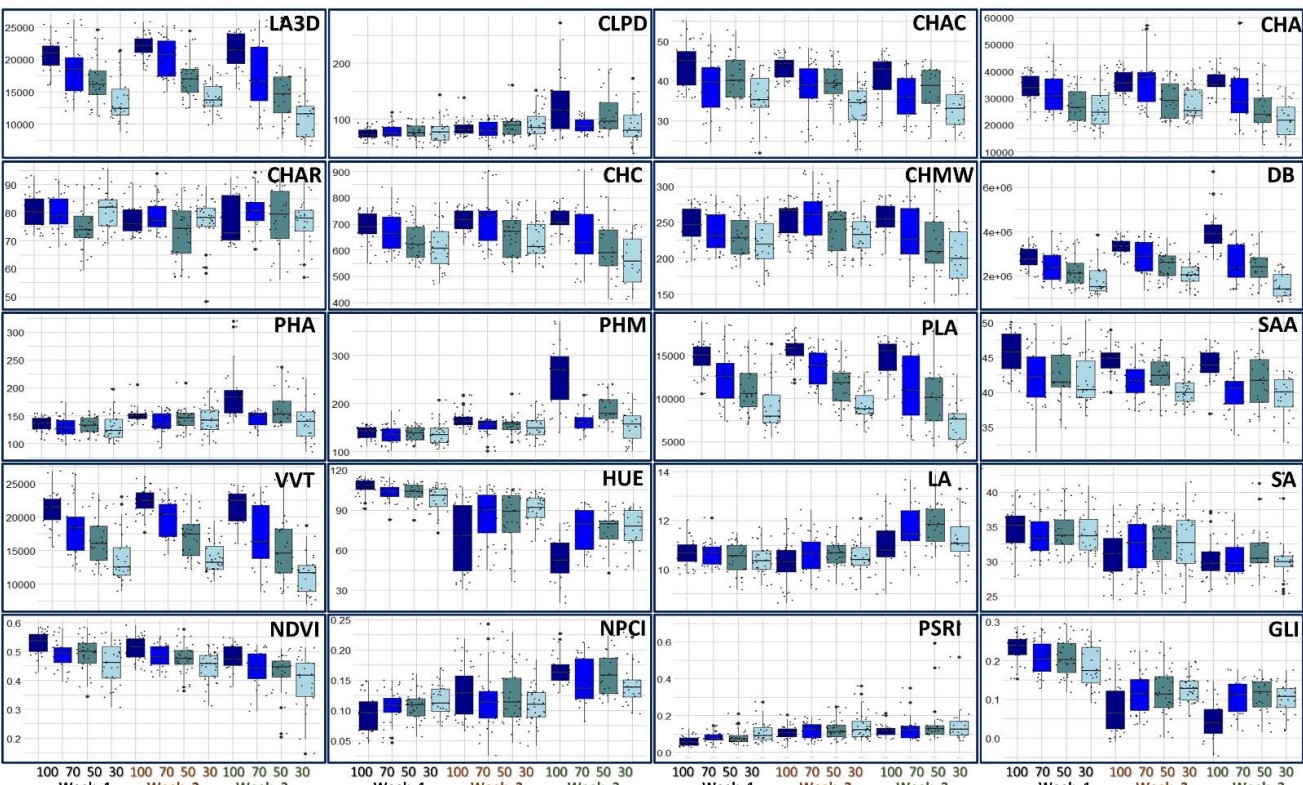

**Figure 2.** Boxplots showing median values and quartiles for the different treatments (100 = control; 70 = 30% less water than the control; 50 = 50% less water than the control; 30 = 70% less water than the control) across three weeks. Trait acronyms: LA3D: Three-Dimensional Leaf Area; CLPD: Canopy Light Penetration Depth; CHAC: Convex Hull Area Coverage; CHA: Convex Hull Area; CHAR: Convex Hull Aspect Ratio; CHC: Convex Hull Circumference; CHMW: Convex Hull Maximum Width; DB: Digital Biomass; PHA: Plant Height Averaged; PHM: Plant Height Max PLA: Projected Leaf Area; SAA: Surface Angle Average; VVT: Voxel Volume Total; HUE: Hue Average; LA: Lightness Average; SA: Saturation Average; NDVI: NDVI Average; NPCI: NPCI Average; PSRI: PSRI Average; GLI: GLI Average.

The Canopy Light Penetration Depth tended to increase in the first two weeks of cultivation with the occurrence of water stress, due to the loss of turgor in the leaves that caused wilting and therefore greater light penetrability. During the third week, we also observed an increase in the Canopy Light Penetration Depth in the control. This is linked to the lowering of the leaves resulting in weight gain following the growth of the plant. The effect of growing is shown by biomass and plant height traits that reached the maximum values during the last week in the control and to a minor extent in the groups with 70% and 50% water-holding capacity. Convex hull traits were also affected by the application of stresses. These parameters provide an estimation of the spread of leaves, contributing to the whole plant size that is reduced by lowering irrigation [15]. Projected Leaf Area and Voxel Volume Total were the other two traits proportionally affected by water stress. These two parameters measure the Projected Leaf Area on a 2D plane and the leaf volume of plants, respectively, thus decreasing significantly when the plant is not in optimal growth conditions. We also observed changes in color parameters during the growth cycle.

We observed a general decrease in hue (more yellowish leaves) and an increase in lightness in all plots from the first to the third week. Color parameters are related to the content of pigments, which changes during the growth cycle. It has been observed that water deficit would not impact the lightness and hue of rocket leaves [16]. Therefore, the discoloration of the leaves from the first to the third week is probably caused by the degradation of chlorophyll in response to a nutrient shortage brought on by the lack of

fertilization [17]. This is also indicated by the GLI, which consistently decreased from the first to the third week. PlantEye F500 enables the automatic computation of vegetative indices that represent the health of the plants. We observed a consistent decrease in the NDVI as the intensity of the stress increased, as well as an increase in PSRI. These are the two main indicators of vegetation development and senescence in crops [18].

Table 3 reports the significance of the differences among the means of the treatments for the traits studied in the different weeks. Overall, the traits showed statistically significant differences over the 3 weeks of this study, with a greater number of significant differences observed in the third week. This could be due to the intensification of the effects of stress in the last week.

**Table 3.** Significance of traits for the 3 weeks of this study.

|  | **Week 1** | **Week 2** | **Week 3** |
|---|---|---|---|
| LA3D | *** | *** | *** |
| CLPD | NS | NS | ** |
| CHAC | *** | *** | *** |
| CHA | *** | *** | *** |
| CHAR | ** | * | NS |
| CHC | *** | *** | *** |
| CHMW | ** | *** | *** |
| DB | *** | *** | *** |
| PHA | NS | NS | *** |
| PHM | NS | ** | *** |
| PLA | *** | *** | *** |
| SAA | *** | *** | *** |
| VVT | *** | *** | *** |
| HUE | *** | *** | *** |
| LA | NS | * | * |
| SA | NS | NS | NS |
| NDVI | *** | *** | *** |
| NPCI | * | NS | * |
| PSRI | *** | NS | NS |
| GLI | ** | *** | *** |

$p < 0.001$ = ***; $p < 0.01$ = **; $p < 0.05$ = *; NS = not significant.

Only for Convex Hull Aspect Ratio, Saturation Average, and Plant Senescence Reflection Index were no significant differences observed during the last week. While the first two traits may not have been affected by stress in the cultivation interval considered, the last had an unexpected trend that requires further investigation to fully understand its effectiveness over the time frame. We also calculated the difference between treatments and the control across the three weeks for the 20 traits assayed (Table 4).

Interestingly, we observed how the differences between the control and treatments, and in particular for the traits linked to the morphology of the plant, intensified in the last week of evaluation, denoting the impact of the effect of the stress applied. The results highlight the feasibility of the high-throughput phenotyping platform to detect differences between different drought stress levels, further suggesting that it is possible to maintain adequate performances with a 30% reduction in water.

The correlation among traits has been calculated for each treatment considering a significance threshold of $p < 0.05$ using the Pearson coefficient. The correlogram within the control condition is reported in Figure 3a, while panels 3b, 3c, and 3d represent the different water deficits applied.

**Table 4.** Significant differences between three water deficit treatments with respect to the control condition according to Dunnett's test.

| | Week 1 | | | Week 2 | | | Week 3 | | |
|---|---|---|---|---|---|---|---|---|---|
| | 70 | 50 | 30 | 70 | 50 | 30 | 70 | 50 | 30 |
| LA3D | * | *** | *** | * | *** | *** | ** | *** | *** |
| CLPD | NS | NS | NS | NS | NS | NS | ** | NS | ** |
| CHAC | ** | NS | *** | *** | ** | *** | ** | NS | *** |
| CHA | NS | *** | *** | NS | *** | *** | NS | *** | *** |
| CHAR | NS | ** | NS | NS | * | NS | NS | NS | NS |
| CHC | NS | ** | *** | NS | ** | *** | NS | *** | *** |
| CHMW | NS | NS | ** | NS | NS | ** | NS | ** | *** |
| DB | NS | ** | *** | ** | *** | *** | *** | *** | *** |
| PHA | NS | NS | NS | * | NS | NS | *** | * | *** |
| PHM | NS | NS | NS | ** | * | ** | *** | *** | *** |
| PLA | ** | *** | *** | *** | *** | *** | *** | *** | *** |
| SAA | *** | ** | *** | *** | *** | *** | *** | * | *** |
| VVT | ** | *** | *** | *** | *** | *** | *** | *** | *** |
| HUE | NS | NS | *** | ** | ** | *** | *** | *** | *** |
| LA | NS | NS | NS | * | * | NS | NS | * | NS |
| SA | NS | NS | NS | NS | NS | NS | NS | NS | NS |
| NDVI | * | * | *** | * | ** | *** | NS | * | *** |
| NPCI | NS | NS | ** | NS | NS | NS | * | NS | * |
| PSRI | NS | NS | *** | NS | NS | * | NS | NS | NS |
| GLI | NS | NS | *** | ** | ** | *** | *** | *** | *** |

$p < 0.001$ = ***; $p < 0.01$ = **; $p < 0.05$ = *; NS = not significant.

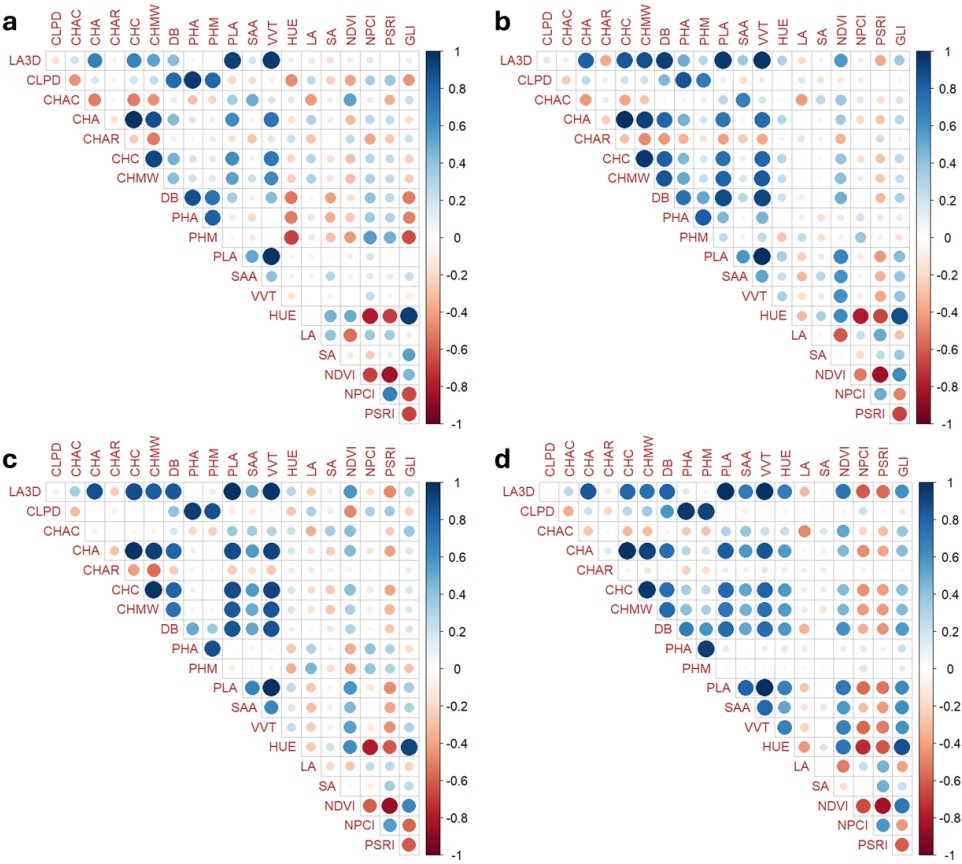

**Figure 3.** Correlations between morphological parameters, color measures, and vegetative indices for the different treatments. (**a**) Control = 100% water; (**b**) 70 = 30% less water than the control; (**c**) 50 = 50% less water than the control; (**d**) 30 = 70% less water than the control. The Pearson coefficient with a significance threshold of $p < 0.05$ was considered. Color intensity and dots size are both directly proportional to the coefficients. According to the scale on the right, blue and red colors correspond to positive and negative correlations, respectively. The full name of each trait abbreviation can be found in Table 1.

The correlograms suggest the same patterns in all trials with an increasing number of strong correlations in the 50 and 30 stress conditions. Interestingly, the correlations between the LA3D and GLI, as well as between the HUE and LA, constantly increased from the control to the maximum reduction in water intake. The NCPI and PSRI exhibited the strongest negative correlations with all morphological and color parameters in severe stress conditions. We also observed how plant maximum height was strongly correlated with the color spectrum characteristics and vegetative indices in the control (Figure 3a), and vice versa in the stress conditions, closer correlations occurred with the other morphological characteristics (Figure 3d). The principal component analyses (PCAs) in the first two dimensions explained 63.8%, 52.1%, and 63.5% of the total variation in the first, second, and third weeks of cultivation, respectively (Figure 4).

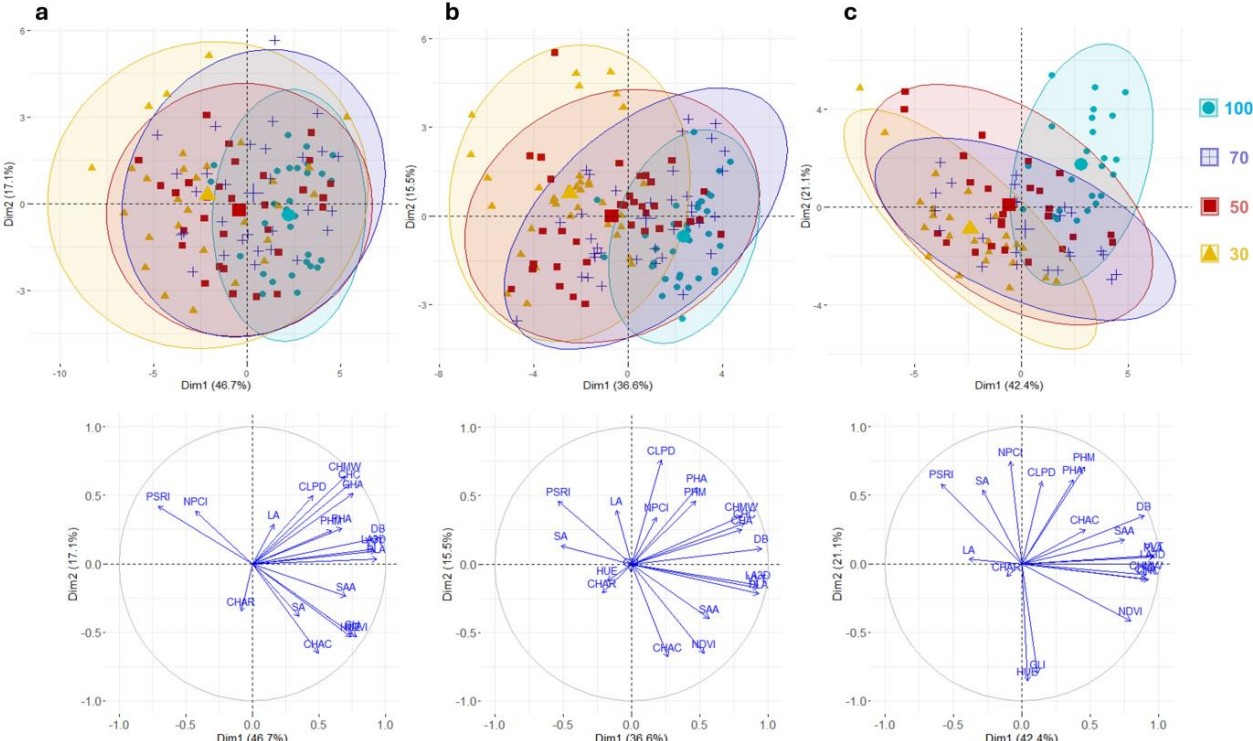

**Figure 4.** Principal component analyses. Loading plot of the first (PC1) and second (PC2) principal components showing the variation for 20 traits scored across three weeks for the different drought stress treatments. (**a**) Week 1; (**b**) week 2; (**c**) week 3. Colored ellipses group measures for each treatment with a 95% confidence interval. The legend indicates different treatments: 100 = control; 70 = 30% less water than the control; 50 = 50% less water than the control; 30 = 70% less water than the control. On the bottom, a distribution of the traits scored on the PCA biplot is displayed. The direction and distance from the center of the biplot indicate how each OTU contributes to the first two components.

Consistently, we observed separation between the control and maximum irrigation reduction across the three periods, while the 70 and 50 stress trials tended to show fewer differences. The PCA evidenced how, in the first two weeks between the control and 30% water deficit, there were slight differences, thus highlighting how the cultivar of rocket salad assessed may tolerate mild drought stresses. The PCA also showed how the performance of most traits was reduced to the occurrence of stress, thus highlighting the PSRI and NDVI as the main discriminators between the control and the three stress trials.

## 4. Conclusions

The brief report presented here, despite being relative to a single accession, is highly informative, providing a comprehensive overview of the morpho-physiological changes in wild rocket salad under drought stress. This confirms that the PlantEye F500 is powerful

enough to dissect the morphological and physiological mechanisms of plants under water stress. The efficiency of the PlantEye F500, combined with the controlled cultivation system, allowed us to minimize any unwanted sources of variability. We observed how severe drought stress may impact plant growth if kept for a long time interval, while a moderate water deficit has minor effects. Our findings highlight that, in rocket salad, it is possible to maintain adequate performance by reducing the irrigation by over 30%. This, adopted on a large scale, would allow a considerable saving of water, which is one of the main targets to achieve in agriculture. The approach described here can be broadened to investigate the performance of cultivars in combination with additional abiotic and biotic stresses. Furthermore, as wild rocket salad is a crop with multiple harvests, it is possible to investigate the effect of stress across multiple cultivation periods. In addition, the implementation of transcriptomics and metabolomics analysis would give us the opportunity to deeply investigate the mechanisms of drought stress tolerance in wild rocket salad in combination with multispectral data.

**Author Contributions:** P.T. and C.P. conceived the project, C.P. set up the experimental plan, A.C. and C.V. performed the experiments, P.T. and A.V. managed the phenotyping platform, P.T. drafted the manuscript. All authors have read and agreed to the published version of the manuscript.

**Funding:** The authors are grateful for the funding support of the Italian Ministry of University and Research (MUR), project "Conservabilità, qualità e sicurezza dei prodotti ortofrutticoli ad alto contenuto di servizio—ARS01_00640—POFACS", D.D. 1211/2020 and 1104/2021; the Agritech National Research Center that received funding from the European Union Next-GenerationEU (PIANO NAZIONALE DI RIPRESA E RESILIENZA (PNRR)—MISSIONE 4 COMPONENTE 2, INVESTIMENTO 1.4—D.D. 1032 17/06/2022, CN00000022); and the Italian Ministry of Agriculture, Food Sovereignty and Forests (MASAF), project RGV-FAO 2023-25.

**Data Availability Statement:** The data described in this article can be freely and openly accessed on [Figshare] under [https://doi.org/10.6084/m9.figshare.25201160, accessed on 6 May 2024; https://doi.org/10.6084/m9.figshare.25201172, accessed on 6 May 2024].

**Conflicts of Interest:** The authors declare that the research was conducted in the absence of any commercial or financial relationships that could be construed as a potential conflict of interest. The funders had no role in the design of the study; in the collection, analyses, or interpretation of data; in the writing of the manuscript; or in the decision to publish the results.

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
