# Peer review of "Precision Phenotyping of Wild Rocket (Diplotaxis tenuifolia) to Determine Morpho-Physiological Responses under Increasing Drought Stress Levels Using the PlantEye Multispectral 3D System"

_horticulturae, doi:10.3390/horticulturae10050496_

Round 1

Reviewer 1 Report

Comments and Suggestions for Authors

I read with interest the manuscript entitled “Precision phenotyping in wild rocket salad (Diplotaxis tenuifolia) to determine morphophysiological responses under increasing drought stress levels”. This study, describes how the advanced phenotyping platform precisely assesses changes in plant architecture and growth parameters of wild rocket salad (Diplotaxis tenuifolia L. [DC.]) under drought stress conditions. Therefore, the manuscript needs some adjustments so that it can then be forwarded to the publication process. The manuscript has the potential for publication in the journal Horticulturae and needs the following adjustments:

TITLE

Is “Rocket salad” the popular name of the species? Or is it just Rocket? To review.

ABSTRACT

Replace keywords that are repeated in the title.

INTRODUCTION

- Divide the paragraph. Despite being a brief communication, it is necessary to divide the information to improve understanding of the ideas in the text. To review.

- Add hypotheses before the objectives, in the last paragraph.

- Add the objective in the last paragraph.

MATERIAL AND METHODS

- Same suggestion as the Introduction. Insert more information and divide paragraphs.

- Do you have a more detailed photo of this phenotyping system used?

- The photo in the article shows the distribution of the plants. More photos of the system would be interesting. From what we can understand, the system used is an innovation that there are few studies on. That's right?

- What statistical analyses are used? Only the program used was mentioned the R program.

RESULTS AND DISCUSSION

- I suggest removing the lines at the bottom of the figure. Leave the background white. Only the box plot was made, how was the comparison between treatments and weeks? Wouldn't it be a factorial scheme? Was any mean comparison test carried out? It is only possible to confirm whether there was a difference between treatments with a mean comparison test. Review this.

- In Figure 4, increase the size of the dice scatter symbols. It is not possible to view this way.

CONCLUSION

Was PlantEye500 used for the first time in this study? I'm curious to know.

Author Response

Dear Reviewer, thank you for the attention to our work and your valuable review comments. We are grateful for the opportunity to include changes in the revised manuscript draft. Below point-by-point answer to all raised concers; all changes have been incorporated in blue font  the revised manuscript

TITLE

Is “Rocket salad” the popular name of the species? Or is it just Rocket? To review.

R: Thanks for notifying this, wild rocket is more appropriate, since rocket salad refers also to other species (e.g.  the cultivated species Eruca vesicaria. We removed the word “salad”.

ABSTRACT

Replace keywords that are repeated in the title.

R: We removed and replaced the repeated keywords.

INTRODUCTION

- Divide the paragraph. Despite being a brief communication, it is necessary to divide the information to improve understanding of the ideas in the text. To review.

- Add hypotheses before the objectives, in the last paragraph.

- Add the objective in the last paragraph.

R: Done, we added a part 64-71 to better explain the hypothesis and objectives of this report.

MATERIAL AND METHODS

- Same suggestion as the Introduction. Insert more information and divide paragraphs.

R: The material and method section has been divided in three paragraph focused on: experimental design, phenotypic measures and data analysis. More information have been provided for the latter one.

- Do you have a more detailed photo of this phenotyping system used?

R: Yes we replaced Figure 1, adding another combined photo showing more details of the PlantEye 3D multiscanner and the phenotyping system.

- The photo in the article shows the distribution of the plants. More photos of the system would be interesting. From what we can understand, the system used is an innovation that there are few studies on. That's right?

R: we confirm, the layout of the phenotyping platform is innovative since have been built according to our preferences. We add a better detailed photo (see the comment above)

- What statistical analyses are used? Only the program used was mentioned the R program.

R: Better details have been provided in the method section.

RESULTS AND DISCUSSION

- I suggest removing the lines at the bottom of the figure. Leave the background white. Only the box plot was made, how was the comparison between treatments and weeks? Wouldn't it be a factorial scheme? Was any mean comparison test carried out? It is only possible to confirm whether there was a difference between treatments with a mean comparison test. Review this.

R: We provide two additional tables with more details on statistical analysis. Table 3 reports the significance of traits across the four irrigation conditions in each week. Here it is possible to see how the significance of some traits increases in the second or third week, better explaining the effects of the 3 treatments, while other traits are not affected by differences across the growing cycle. Table 4 report the  Dunnett’s test of treatments vs the control in the 3 independent weeks of analysis. In most instances is possible to see the significativity of effect of the severe drought stress

- In Figure 4, increase the size of the dice scatter symbols. It is not possible to view this way.

R: We provide an improved Figure 4 with increased dice scatter symbols

CONCLUSION

Was PlantEye500 used for the first time in this study? I'm curious to know

R: Yes in wild rocket, we stated this in the introduction (L69-71). The conclusion section has been improved

Reviewer 2 Report

Comments and Suggestions for Authors

This manuscript reports a controlled study on the responses of wild rocket salad to different levels of water stress. The authors recorded canopy reflectance signatures using a commercial multispectral scanners and linked the measured digital signatures with changes in plant biomass, leaf area, growth and plant architectural traits daily for three weeks of cultivation, Their results clearly show the feasibility of using the scanner to perform high throughput phenotyping. The design was appropriate and the manuscript is generally well written. It is therefore suitable for publication after some minor revisions.

Abstract: Please show some quantitative key results (i.e. statistically significant) to determine the feasibility of high throughput phenotyping to differentiate drought stress levels. The last sentence is incomplete, please change to This study provided meaningful ...

Introduction: Please provide the hypothesis and objectives of the study.

M&M: Not sure why just peat was used as the potting medium and not mixed with sand, perlite and/or soil. It is best to provide Hoagland solutions at some point to avoid confounding the effect of plant nutrient deficiency with the targeted water stress. 

Spectral reflectance of the instrument described as the passing wavelength band, and needs to describe the peak (midpoint of the passing band?)

Methods for calculating spectral indices, such NPCI and PSRI should be given. Some of the measurement units reported do not make sense, e.g. leaf area in mm2 and biomass in mm.

Was the position of the potted plants rotated during growth and measurement to avoid position-induced changes in growth?

How long did it take to scan all the plants each time? At what time of day were measurements taken relative to pot moisture content reaching the target stress levels?

R&D: The data reported in Fig. 2 appeared to be descriptive. Are there any statistically significant differences between treatments?

The significance of the results found in this study could be further explained with references to the literature and key points of the study can be highlighted.

Conclusions: Please briefly speculate on how the findings could be used to develop best irrigation management strategies and/or cultivar improvement.

Author Response

We would like to thank you for the attention to our work. All revisions have been carried out according to suggestions and have been incorporated in blue font in the revised manuscript. We provide point-by-point answer to all raised comments.

Abstract: Please show some quantitative key results (i.e. statistically significant) to determine the feasibility of high throughput phenotyping to differentiate drought stress levels. The last sentence is incomplete, please change to This study provided meaningful ...

R: We add few more details of results in the abstract (L29-32)

Introduction: Please provide the hypothesis and objectives of the study.

R: Done, we added a part 64-71 to better explain the hypothesis and objectives of this report.

M&M: Not sure why just peat was used as the potting medium and not mixed with sand, perlite and/or soil. It is best to provide Hoagland solutions at some point to avoid confounding the effect of plant nutrient deficiency with the targeted water stress. 

R: Thanks for this useful advice that we will adopt in further studies and other crops. In this case we monitored the cycle for 3 weeks, growing all plants in the same conditions. Despite any nutrient deficiency may occur, this is limited compared to the water stress conditions. Therefore, any occurring nutritional stress would not impact the analysis.

Spectral reflectance of the instrument described as the passing wavelength band, and needs to describe the peak (midpoint of the passing band?)

R: In the method section we included more details of the peak wavelength and the spectral half width for each channel see L101-105.

Methods for calculating spectral indices, such NPCI and PSRI should be given.

R: we added this information as a subscript in Table 1

Some of the measurement units reported do not make sense, e.g. leaf area in mm2 and biomass in mm.

R:We checked, there was a mistake for biomass for which the unit is mm3 (not mm2), for leaf area, mm2 is correct dire qualcosa su NDVI

Was the position of the potted plants rotated during growth and measurement to avoid position-induced changes in growth?

R: Plants are fixed, the scanner mover over the plant allowing the to acquire all measure and be reconstructed the entire canopy. This was described in the method L97-98

How long did it take to scan all the plants each time? At what time of day were measurements taken relative to pot moisture content reaching the target stress levels?

R: we report these information in the method part at L89 and L 99-100. Moisture pot content was measured every morning at 10 a.m. Plants were scanned daily at 11:30. The scanner took 20 minutes to complete the analysis.

R&D: The data reported in Fig. 2 appeared to be descriptive. Are there any statistically significant differences between treatments?

R: We provide two additional tables with more details on statistical analysis. Table 3 reports the significance of traits across the four irrigation conditions in each week. Here it is possible to see how the significance of some traits increases in the second or third week, better explaining the effects of the 3 treatments, while other traits are not affected by differences across the growing cycle. Table 4 report the  Dunnett’s test of treatments vs the control in the 3 independent weeks of analysis. In most instances is possible to see the significativity of effect of the severe drought stress

The significance of the results found in this study could be further explained with references to the literature and key points of the study can be highlighted.

R: We expanded results and discussion and highlight main findings in the conclusion section. We would like to remark that this is a brief report, showing the potentiality of this phenotyping system in wall rocket. Results need to be deeply explored in further investigations

Conclusions: Please briefly speculate on how the findings could be used to develop best irrigation management strategies and/or cultivar improvement.

R: we improved the conclusion section

Round 2

Reviewer 1 Report

Comments and Suggestions for Authors

Dear,

The authors made the corrections. The article has the potential to be published.